# Induced Remodelling of Astrocytes In Vitro and In Vivo by Manipulation of Astrocytic RhoA Activity

**DOI:** 10.3390/cells12020331

**Published:** 2023-01-15

**Authors:** Cátia Domingos, Franziska E. Müller, Stefan Passlick, Dagmar Wachten, Evgeni Ponimaskin, Martin K. Schwarz, Susanne Schoch, André Zeug, Christian Henneberger

**Affiliations:** 1Institute of Cellular Neurosciences, Medical Faculty, University of Bonn, 53127 Bonn, Germany; 2Cellular Neurophysiology, Hannover Medical School, 30625 Hannover, Germany; 3Institute of Innate Immunity, Medical Faculty, University of Bonn, 53127 Bonn, Germany; 4Institute of Experimental Epileptology and Cognition Research (EECR), Medical Faculty, University of Bonn, 53127 Bonn, Germany; 5Institute of Neuropathology, University of Bonn Medical School, 53127 Bonn, Germany; 6German Center for Neurodegenerative Diseases (DZNE), 53127 Bonn, Germany

**Keywords:** astrocytes, morphology, RhoA, cytoskeleton

## Abstract

Structural changes of astrocytes and their perisynaptic processes occur in response to various physiological and pathophysiological stimuli. They are thought to profoundly affect synaptic signalling and neuron-astrocyte communication. Understanding the causal relationship between astrocyte morphology changes and their functional consequences requires experimental tools to selectively manipulate astrocyte morphology. Previous studies indicate that RhoA-related signalling can play a major role in controlling astrocyte morphology, but the direct effect of increased RhoA activity has not been documented in vitro and in vivo. Therefore, we established a viral approach to manipulate astrocytic RhoA activity. We tested if and how overexpression of wild-type RhoA, of a constitutively active RhoA mutant (RhoA-CA), and of a dominant-negative RhoA variant changes the morphology of cultured astrocytes. We found that astrocytic expression of RhoA-CA induced robust cytoskeletal changes and a withdrawal of processes in cultured astrocytes. In contrast, overexpression of other RhoA variants led to more variable changes of astrocyte morphology. These induced morphology changes were reproduced in astrocytes of the hippocampus in vivo. Importantly, astrocytic overexpression of RhoA-CA did not alter the branching pattern of larger GFAP-positive processes of astrocytes. This indicates that a prolonged increase of astrocytic RhoA activity leads to a distinct morphological phenotype in vitro and in vivo, which is characterized by an isolated reduction of fine peripheral astrocyte processes in vivo. At the same time, we identified a promising experimental approach for investigating the functional consequences of astrocyte morphology changes.

## 1. Introduction

Astrocytes have a diverse, complex, and dynamic cellular structure. Similar to neurons, they display a considerable morphological heterogeneity across brain regions [1] and between layers of the same brain region [2,3]. Their morphology evolves during postnatal development as observed, for instance, in the hippocampus [3,4]. In addition to a ‘backbone’ of large GFAP-positive processes, astrocytes reach out with fine processes towards synapses, where they engage in bidirectional neuron-astrocyte signalling [5,6]. The full geometry of these perisynaptic astrocyte processes (PAPs) is incompletely resolved by diffraction-limited microscopy [7], which complicates research. As a consequence, new morphological features such as ring-like astrocyte structures are still being discovered [8,9].

At the same time, the morphology of astrocytes and especially that of their PAPs is not static. They have been shown to be motile [10,11], and their morphology is rapidly altered by the induction of synaptic long-term potentiation (LTP) [12,13,14,15]. Astrocyte remodelling was also observed in various physiologically relevant contexts. For instance, lactation is accompanied by a re-arrangement of neuronal and astrocytic membranes in the supraoptic nucleus [16], and PAPs undergo circadian structural changes in the hippocampus [17]. Similarly, the spatial arrangement of PAPs and excitatory synapses is correlated with the sleep–wake cycle in the prefrontal cortex [18], altered by sensory stimulation in the barrel cortex [19], and affected by caloric restriction in the hippocampus [20]. Such structural changes of PAPs at excitatory synapses are relevant for synapse function because studies have linked, for example, PAP structure and its changes to glutamate uptake and spread as well as to extra- and presynaptic glutamate receptor activation [13,21,22].

These observations indicate that the morphology of astrocytes must be finely tuned and adapted, which requires a balanced activity of signalling cascades to maintain and modify the astrocyte cytoskeleton. This raises the question which intracellular signalling cascades are relevant for astrocyte remodelling in complex scenarios, such as lactation and LTP induction, and if isolated activation of these cascades reproduces morphological and functional outcomes. 

Small GTPases of the Rho family are potent candidate modulators of astrocyte morphology [23], and RhoA activity has been linked to the inhibition of astrocyte process outgrowth and stellation [24,25,26,27,28]. Our recent finding, that astrocytic serotonin receptors (5-HT_4_R) signal via Gα_13_-RhoA and Gα_S_ to induce actin-remodelling and astrocyte morphology changes in cultured astrocytes [29], emphasizes that astrocytic RhoA could be a powerful modulator of astrocyte morphology. However, it remains to be established if astrocytic RhoA activity on its own is sufficient to consistently modify astrocyte morphology in vitro and in vivo. 

We therefore designed and tested a viral approach that is suitable for manipulating astrocyte RhoA activity selectively. We tested several RhoA variants, which have previously been shown to alter RhoA signalling [30,31], for their ability to change astrocyte morphology: wild-type RhoA (RhoA-WT), a dominant-negative RhoA (RhoA-DN), and a constitutively active variant (RhoA-CA). These were also chosen to test if bi-directional changes of astrocyte morphology could be induced (e.g., RhoA-CA vs. RhoA-DN).

Our results reveal that a prolonged increase of astrocytic RhoA activity indeed modifies astrocyte morphology in vitro and in vivo by inducing an astrocyte process withdrawal in vitro and, similarly, a reduction of smaller astrocytic processes in vivo. Among the tested RhoA variants, RhoA-CA induced the most prominent and consistent astrocyte morphology changes, which makes it a promising tool for exploring the functional consequences of induced astrocyte process withdrawal in future studies.

## 2. Material and Methods

### 2.1. Animals

Transgenic mice expressing EGFP under a GFAP promotor [GFAP-EGFP, [32]] of either gender (FVB background) and C57Bl/6J mice were used in this study. Their age is indicated for individual experiments and throughout the methods. Animals were housed under 12 h light/dark conditions with food and water ad libitum. All animal procedures were conducted in accordance with the regulations of the European Commission directive 2010/63/EU and all relevant national and institutional guidelines and requirements. All procedures were approved by the Landesamt für Natur, Umwelt und Verbraucherschutz Nordrhein-Westfalen (LANUV, Recklinghausen, Germany) where required. 

### 2.2. Design and Production of Recombinant Adeno-Associated Viruses (rAAVs)

Plasmids for RhoA overexpression were originally obtained from Dr. A. Woehler (MDC, Berlin, Germany). The constitutively active variant RhoA-CA contains the G14V mutation and the dominant-negative RhoA-DN the T19N mutation compared to the wild-type (WT) sequence [30,31]. RhoA-variants were cloned into the pAAV-mGFAP-TurboFP650 vector, which was further used as a control. Briefly, a megaprimer PCR was used to add a HindIII restriction site at the 3′ end of a mCherry-RhoA insert. Insert and vector were cut with restriction enzymes BamHI and HindIII and combined using T4 ligase. The resulting pAAV-mGFAP-mCherry-RhoA variant plasmids were verified by sequencing. For in vitro experiments, adeno-associated viruses (AAVs) were produced using the AAV-DJ Packaging System from Cell Biolabs (San Diego, CA, USA). HEK293FT cells were grown to about 80% confluency and transfected with plasmids containing the gene of interest along with a pHelper plasmid as well as a pAAV-DJ plasmid using 1 mg/mL polyethylenimine. AAVs were harvested after three days by three freezing and thawing cycles in lysis buffer followed by a benzonase digest. AAVs were subsequently concentrated using Amicon Ultra Centrifugal Filter (Millipore, Billerica, MA, USA) columns with a cut-off of 10 kDa. For titer determination, virus capsids were digested using proteinase K, and virus DNA was analysed using quantitative real-time polymerase chain reaction (qRT-PCR) with primers against the WPRE element (fw 5′-CCTGGTTGCTGTCTCTTTATGAGG-3′; rev 5′-TGACAGGTGGTGGCAATGC-3′; probe 5′-/6-FAM/CGTTGTCAGGCAACGTGGCGTGGTG/TAMRA/-3′; Sigma). For viral gene copy number calculation per μL, the relative standard curve method was used.

For in vivo experiments, mCherry was replaced by the improved red fluorescent protein tdTomato, and the murine GFAP promoter was exchanged for a shorter human GFAP promoter with enhanced specificity in the hippocampal CA1 target region [33]. In a first step, the fluorescent protein was exchanged using EcoRI restriction sites added to a tdTomato sequence (pME-tdTomato no stop, Addgene, Watertown, MA, USA, #82404). The 5′ EcoRI restriction site in pAAV-mGFAP-tdTomato-RhoA variants was replaced by BamHI using overlap extension PCR. tdTomato-RhoA variants were then cut out using restriction sites BamHI and HindIII and inserted into a pAAV-hGFAP vector (replacing iGluSnFR from pAAV1-GFAP-iGluSnFr-WPRE-SV40) [33] with corresponding restriction sites (plasmid obtained from PennCore, Addgene, #98930). The resulting plasmids were verified by sequencing. AAV1-hGFAP-tdTomato-RhoA variants were produced in HEK293T cells, harvested, and purified as previously described in detail [3,34,35]. Briefly, the pAAV1-hGFAP-tdTomato-RhoA variant-WPRE-SV40 plasmid was co-expressed with the helper plasmids pRV1, pH21, and pFΔ6 in HEK283T cells and harvested ~48 h later. Next, cells were lysed, and virus particles were purified by HiTrapTN heparin columns and concentrated with Amicon Ultra Centrifuge Filters to a final stock of 500 μL.

### 2.3. Stereotactic Virus Injections

Viral particles (see above) were injected bilaterally into the dorsal hippocampus of either GFAP-EGFP or Thy1-YFP mice, as previously described [3,21]. Briefly, 4–6-week-old mice were anesthetized by intraperitoneal (i.p.) injection of Fentanyl/Midazolam/Medetomidin (0.05/5.0/0.5 mg/kg body weight) and stereotactically injected (coordinates for the dorsal hippocampus, relative to bregma: anterior −1.8 mm, lateral +/− 1.6 mm, ventral −1.6 mm) with 1 µL of the AAV1-hGFAP-tdTomato-RhoA variants (virus titer ~10^9^ viral particles per µL, injection speed 100 nL/min). Anaesthesia was terminated by Naloxon/Flumazenil/Atipamezol (1.2/0.5/2.5 mg/kg body weight, i.p. injection). To ensure analgesia, carprofen (5 mg/kg in NaCl, injection volume 0.1 mL/20 g body weight, Rimadyl 50 mg/mL, Zoetis, Parsippany, NJ, USA) was injected subcutaneously 30 min before and 24 h after surgery. Animals were sacrificed 2–4 weeks after virus injection.

### 2.4. Preparation and Transduction of Dissociated Cultures

Primary astrocyte cell cultures were prepared as previously described [29]. Briefly, whole brains were obtained from mice at postnatal day 0–3, and cells from dissociated hippocampi were seeded at a density of 5 × 10^4^ cells per 12 mm glass coverslip in 500 µL plating medium (49 mL Minimal Essential Medium, 1 mL B-27, 500 µL of 200 mM glutamine, 500 µL of 100 mM sodium pyruvate, 5 u/mL penicillin, 5 mg/mL streptomycin). On day in vitro (DIV) 3, the entire plating medium was replaced with 1 mL maintenance medium (49 mL Neurobasal-A medium, 1 mL B-27, 500 µL of 200 mM glutamine, 5 u/mL penicillin, 5 mg/mL streptomycin). Cell cultures were maintained at 37 °C in a humidified incubator in a 5% CO_2_ atmosphere until they were used for experiments. Half of the medium was exchanged on DIV11 with maintenance medium prior to viral transduction of the cells. During microscopy, cells were kept in a balanced salt solution containing 115 mM NaCl, 5.4 mM KCl, 1 mM MgCl_2_, 2 mM CaCl_2_, and 20 mM HEPES, adjusted to pH 7.4 and 290 mOsm with glucose. Astrocytes were used for experiments on DIV14-17. 

The purity and composition of the cultures were documented previously [29]. They contained GFAP-expressing cells (i.e., astrocytes, 85%), βIII-tubulin–expressing cells (i.e., neurons, 2.8%), and cells negative for both markers (12%). Please see [29] for further details. The culturing protocol was chosen to obtain low to medium density cultures of astrocytes with stellate morphologies. For alternative culturing protocols and their effect on astrocyte morphology, see [36,37,38].

### 2.5. Immunohistochemistry

For labelling of filamentous (F) and globular (G) actin, cultured astrocytes were fixed for 10 min with 4% paraformaldehyde (PFA), permeabilized in 100% acetone for 3 min, and then incubated with DNaseI linked to Alexa Fluor 488 (9 µg/mL, #D12371, ThermoFisher Scientific, Waltham, MA, USA) and Phalloidin-Atto 594 (7.9 ng/mL, #51927, Sigma, St. Louis, MI, USA) in blocking solution (1% BSA in phosphate-buffered buffer saline, PBS, pH 7.4) for 30 min. Coverslips were mounted on glass slides with Fluoromount-G. 

For immunohistochemistry of virus-injected mice, animals were deeply anesthetized (two to three weeks after virus injection) by intraperitoneal injection of a Ketamine/Xylazin solution and transcardially perfused with ice-cold 4% PFA in PBS. Their brains were then removed and postfixed in PFA at 4 °C overnight (ON). Coronal brain sections of (70 µm thickness) were cut on a vibratome (Leica, VT1200S; Leica Microsystem) and washed 3× for 10 min in PBS. Slices selected for immunostaining were then incubated in a permeabilization and unspecific binding block solution (0.5% Triton X-100, 5% normal goat serum (NGS) in PBS) for 1 to 1.5 hrs at RT. Afterwards, brain slices were incubated with primary antibodies diluted in PBS at 4 °C overnight (ON). Primary antibodies used were chicken anti-GFAP (1:500, Synaptic systems, 173006), mouse anti-NeuN (1:500, Millipore, Billerica, MA, USA, MAB377), mouse anti-GFAP (1:500, Millipore, MAB360), chicken anti-GFP (1:500, Abcam, Boston, MA, USA, ab13970), and rabbit anti-mCherry (1:500, Abcam, ab167453). After 3× 10 min washing in PBS at RT, slices were incubated in secondary antibodies diluted in PBS for 2 to 2.5 h at RT. Secondary antibodies used were goat anti-chicken Alexa Fluor 594 (1:200, Invitrogen, Waltham, MA, USA A11042), goat anti-mouse Alexa Fluor 594 (1:200, Invitrogen, A11032), goat anti-chicken Alexa Fluor 488 (1:500, Invitrogen, A11039), goat anti-rabbit Alexa Fluor 594 (1:500, Invitrogen, A11037), and biotin-SP-AffiniPure Fab fragment goat anti-mouse (1:500, Jackson Immunoresearch, 115-067-003). For simultaneous immunohistochemistry of GFAP, tdTomato and EGFP, GFAP was visualized using the biotin-labelled secondary antibody and a third incubation step with Alexa Fluor 647 conjugated streptavidin (1:200, Johnson Immunoresearch, Ely, United Kingdom, 016-600-084). In some experiments, slices were also incubated with Hoechst 33342 (1:2000, Invitrogen, H3570) in distilled water for 10 min at RT. Finally, slices were washed 3× for 10 min in PBS and mounted with Invitrogen Prolong Diamond Antifade mountant (P36965) and left for curing for at least 24 h at 4 °C. 

### 2.6. Microscopy and Image Analysis

Images of cultured astrocytes were taken using a Zeiss LSM780 microscope with a LD C-Apochromat 40×/1.2 W objective and Zen2013 imaging software (Zeiss, Oberkochen, Germany) in online fingerprinting mode with previously defined spectra for each fluorescent protein and dye obtained from single expression stainings. Image data were analysed using Matlab (Mathworks, Natick, MA, USA). Morphological analysis was done by analysing the shape of the astrocytes after applying a threshold using the triangle method (skewed bi-modality, maximum distance to triangle) [39]. A region of interest (ROI) was used to select a particular astrocyte structure. Morphological parameters were obtained using the DIPimage toolbox (Release 2.9). F- and G-actin analysis was described previously [29]. 

To quantify the cell type specificity of rAAVs with the hGFAP promotor, fixed coronal slices of the dorsal hippocampus were obtained 2–3 weeks after virus injection of 4-week-old FVB mice. They were stained for GFAP or NeuN (see above), and the results were captured on a Leica SP8 confocal microscope using a 40×/1.1 NA objective. An image stack of the dorsal hippocampus was obtained by tile merging (x-y typically ~1800 µm × 1400 µm, 7 z-steps of 4 µm). Transduced cells were identified by their tdTomato fluorescence, and the number of transduced cells in the hippocampus with co-localization of GFAP or NeuN was determined manually using Fiji/ImageJ. 

The analysis of GFAP-positive processes of astrocytes transduced with RhoA-CA and control astrocytes was performed in fixed coronal slices of the dorsal hippocampus. Samples were obtained three weeks after injecting 4–5-week-old GFAP-EGFP mice with the rAAVs hGFAP-tdTomato-RhoA-CA or hGFAP-tdTomato (control). The fluorescence intensity of EGFP and tdTomato was amplified using immunohistochemistry, and GFAP was labelled as explained above. Transduced astrocytes were identified by their tdTomato fluorescence and selected for imaging if endogenous EGFP expression was present and their territory boundaries were well-defined. For analysis, image stacks of entire astrocytes were acquired by confocal imaging (x-y: 96.97 µm × 96.97 µm, pixel size 0.095 µm × 0.095 µm, ~30 z-steps of 1 µm). GFAP process reconstruction and analysis of morphometric parameters was performed using MotiQ [40], which is freely available on GitHub. Briefly, the GFAP-signal of single transduced astrocytes was isolated using the MotiQ cropper. For each image in an image stack, the astrocyte territory was manually delineated using its EGFP fluorescence, and GFAP immunofluorescence outside that territory was discarded. The three-dimensional astrocytic GFAP process tree was then reconstructed using the MotiQ thresholder and MotiQ 3D analyser. Three-dimensional GFAP reconstructions were obtained for each astrocyte, and morphometric parameters were provided by the software (average and cumulative segment length, number of branches and branch points).

The analysis of astrocyte morphology in acute slices was performed as previously described [3]. Acute coronal slices of the dorsal hippocampus were obtained 2–4 weeks after injection of rAAVs (hGFAP-tdTomato, hGFAP-tdTomato-RhoA-WT, hGFAP-tdTomato-RhoA-DN, hGFAP-tdTomato-RhoA-CA) into mice expressing EGFP under a GFAP promotor [GFAP-EGFP, 32] and inspected using 2PE (Coherent Vision S, Coherent, Santa Clara, CA, USA, λ_exc._ = 900 nm) fluorescence microscopy (Olympus FV10MP, Olympus Europe, Hamburg, Germany) as described previously [3,41]. Transduced astrocytes were identified by their tdTomato fluorescence. A single horizontal section of transduced astrocytes also expressing EGFP (λ_exc._ = 900 nm) was obtained, and their EGFP fluorescence was analysed (image size 83.97 µm × 83.97 µm, pixel size 0.041 µm × 0.041 µm, average of 5 frames). The outline of each astrocyte was determined, and its area was calculated (the astrocyte cross section area or territory). Before further analysis, the fluorescence intensity was background-corrected by subtracting the average pixel values of a region devoid of labelled structures in the same image. Then the fraction of tissue volume occupied by astrocytes in that territory (astrocyte volume fraction, VF) was determined by normalizing the average EGFP fluorescence intensity in that territory to the average EGFP fluorescence intensity in a somatic reference region [3,13,42]. The VF distribution for a particular cell was obtained by automatically dividing is territory into a grid-like set of regions of interest of 0.5 µm^2^ and calculating the local VF as described. See [3] for an extensive description and verification of the approach. For each virus-injected animal, 3–5 acute brain slices per hippocampus were typically used. In each of these brain slices, 1–17 astrocytes were analysed. For each condition, ≥3 animals were injected with viruses. 

### 2.7. Statistical Analysis

Numerical data are reported as mean ± s.e.m., with n being the number of samples. For details about Box and Whisker plots, see figure legends. Statistical tests such as ANOVA or Kruskal–Wallis were chosen depending on whether data were normally distributed, which was established by the Shapiro–Wilk test. Appropriate parametric and non-parametric tests were then applied as indicated throughout the text and figure legends using Matlab (Mathworks), Origin (OriginLab, Northampton, MA, USA), Prism (GraphPad Software, Boston, MA, USA), or R. Only two-sided tests were used. *p* represents the level of significance. Significance levels are indicated in figures by asterisks (* *p* < 0.05, ** *p* < 0.01, *** *p* < 0.001) unless stated otherwise. Details about the statistical tests and results can be found in the figures, figure legends, and/or text.

## 3. Results

### 3.1. Increased RhoA Activity Modifies Astrocyte Structure In Vitro

We first tested several RhoA variants for their ability to consistently alter astrocyte morphology in dissociated cultures when being overexpressed. We selected the wild-type RhoA (RhoA-WT), a dominant-negative (RhoA-DN, mutation T19N) and a constitutively active variant (RhoA-CA, mutation G14V), which have previously been shown to modify RhoA signalling [30,31]. These were expressed as fusion proteins with the fluorescent protein mCherry under the control of a mGFAP promotor using recombinant adeno-associated viruses (rAAVs) in primary cultures of dissociated astrocytes. Astrocytes transduced with the far-red fluorescent protein TurboFP650 (rAAV, mGFAP-TurboFP650) served as a control (Figure 1A). Three days after transduction, cells were visualized and pseudo-randomly selected for the analysis of their morphology. We used the area covered by individual cells and their perimeter as two-dimensional and easily obtainable screening parameters and found that RhoA-CA expression significantly reduced both (Figure 1B, top panels). A similar tendency was observed for RhoA-WT, although it did not reach statistical significance. To gain further information about the change of astrocyte morphology, we also analysed for each cell the area and the perimeter of its convex hull (i.e., the smallest polygon fully containing the astrocyte with all inner angels below 180°). For a perfectly convex cell (a round disc for example), the convex hull is identical to the cell shape. In contrast, if a cell has a highly branched morphology, the area of its convex hull (convex area) is much larger than the true cell area. RhoA-CA reduced the convex area significantly (Figure 1B, bottom left panel) to levels comparable to the true area, which indicates that RhoA-CA expression leads to a major collapse of astrocytic processes. Interestingly, RhoA-WT also significantly reduced the convex area, which suggests that astrocytes are less ramified after this manipulation. Similar effects were observed when the perimeters of the convex hulls were analysed (Figure 1B, bottom right panel). Together, these results reveal a robust retraction of astrocytic processes after expression of RhoA-CA.

### 3.2. Increased RhoA Activity Alters Astrocytic Actin Assembly in Vitro

This observed morphology change could be explained by a RhoA-dependent modification of the cytoskeleton. Increased RhoA activity is expected to lead to increased phosphorylation of cofilin, to decreased actin depolymerisation and, in turn, to an increased fraction of filamentous actin [29,43]. Therefore, we next tested if expression of RhoA-CA would shift the balance between globular actin (G-actin) and filamentous actin (F-actin) towards F-actin using an established assay [29]. Indeed, we found that the F-actin fraction, i.e., the ratio of F-actin and the total actin (F-actin/(F-actin + G-actin)), was increased in RhoA-CA expressing astrocytes compared to control conditions (Figure 2A,B). Having confirmed this, we further characterized the astrocyte remodelling induced by RhoA expression by performing a Sholl analysis [44] on isolated cells such as shown in Figure 1A. We found that astrocytic processes were overall shorter (Figure 2C), as predicted from our previous results.

### 3.3. Remodelling of Astrocytes In Vivo by Increased RhoA Activity

We next asked if manipulating RhoA activity affects astrocyte structure in vivo in a similar manner. In addition to establishing the effect in the intact brain, this also provides useful information because culturing conditions are known to have a strong effect on astrocyte morphology (see [36,37,38] and references therein). In preliminary tests, we injected the rAAVs described above into the dorsal CA1 *stratum radiatum* of 4–6-week-old mice sparsely expressing EGFP in astrocytes GFAP-EGFP [32]. Visual inspection of fixed slices from injected hippocampi revealed that many neurons were transduced (not quantified). Therefore, we exchanged the murine GFAP (mGFAP) promotor in the rAAVs by hGFAP (see Material and Methods). We then examined if transducing cultured astrocytes with hGFAP-tdTomato-RhoA-CA rAAVs had the same effect on their morphology compared to hGFAP-tdTomato as in previous experiments (hGFAP-tdTomato, control: n = 132 cells, hGFAP-tdTomato-RhoA-CA, Rho-CA: n = 210 cells, data from four independent experiments). Indeed, the effect was qualitatively the same (cell area, control and RhoA-CA: 3639 ± 137 µm^2^ and 2198 ± 62.5 µm^2^; cell perimeter, control and RhoA-CA: 668 ± 28.2 µm and 341 ± 11.4 µm; convex cell area, control and RhoA-CA: 6521 ± 302 µm^2^ and 3037 ± 114 µm^2^; convex cell perimeter, control and RhoA-CA: 311 ± 6.9 µm and 210 ± 4.1 µm; *p* < 0.001 throughout, Mann–Whitney U-tests).

We then injected these new rAAVs (hGFAP-tdTomato, hGFAP-tdTomato-RhoA-WT/CA/DN) into the dorsal CA1 *stratum radiatum* of 4–6-week-old GFAP-EGFP mice (Figure 3A) and quantified the cell type-specificity of rAAVs using immunohistochemical labelling of neurons (NeuN) and astrocytes (GFAP). We found that the rAAV-mediated expression was almost exclusively detected in astrocytes (Figure 3B,C). 

To analyse astrocyte morphology changes induced by the RhoA variants, we visualized transduced astrocytes 2–4 weeks after virus injection in acute hippocampal slices, because chemical fixation of brain tissue has been reported to alter the structure of small astrocyte processes [45] (Figure 4A). Images of transduced astrocytes (tdTomato-positive) also expressing EGFP were taken using 2PE fluorescence microscopy. The morphology of transduced astrocytes was analysed using their EGFP fluorescence, because a differential subcellular distribution of the expressed tdTomato-RhoA-WT/CA/DN fusion proteins could bias analysis, whereas EGFP is freely diffusing inside the astrocyte cytosol [41]. The morphology of transduced cells was quantified using established indirect measures, because fine details of astrocyte morphology cannot be fully resolved by diffraction-limited microscopy. Specifically, we measured the fraction of tissue volume occupied by astrocytes (volume fraction, VF), which we previously confirmed to be a sensitive indicator of astrocyte remodelling [3,13,21,42]. The volume fraction was determined in a single horizontal section through the astrocyte cell body, which we showed to be representative of the entire astrocyte [3]. We found that all RhoA variants reduced the astrocytic volume fraction, and the effect was strongest for RhoA-CA (Figure 4B). To further characterize this volume fraction change, we analysed the subcellular distribution of volume fractions [3,13,46] by automatically dividing astrocyte cross sections into small rectangular subregions of interest and calculating the astrocyte volume fraction in each. No differences were detected between distributions of control and RhoA-CA–expressing astrocytes for local volume fractions >12.5% (not illustrated, see legend of Figure 4 for statistical analysis), which typically represent regions of interest covering larger astrocyte processes. However, below 12.5%, which corresponds to regions covered by fine astrocytic processes, significant differences were detected (Figure 4C). Local astrocytic volume fractions below 2.5% were observed more often and volume fractions between 5 and 10% less often after RhoA-CA expression (Figure 4C). We obtained qualitatively similar results for RhoA-DN and RhoA-WT expression (not illustrated). This indicates that small astrocytic processes became rarer and/or thinner, which is in line with an astrocyte process retraction [3,13] and with what we observed in culture experiments. Therefore, our experiments reveal that RhoA-CA expression in astrocytes induces a retraction of their small processes in vitro and in vivo.

### 3.4. Unaltered Large Process Structure after Increasing RhoA Activity 

Our VF analysis also indicated that larger astrocyte processes might not be affected by RhoA-CA expression in vivo. This was further investigated by transducing astrocytes in the dorsal hippocampus of GFAP-EGFP mice with control rAAVs or rAAVs expressing RhoA-CA (see above), then labelling the GFAP-positive major branches of astrocytes by immunohistochemistry and reconstructing the tree of GFAP-positive branches of individual astrocytes in 3D using MotiQ [40] (Figure 5A,B). Using the reconstructions, we analysed the number of GFAP-positive segments, their average and cumulative lengths per cell, and the number of branch points. No differences between control and RhoA-CA–expressing astrocytes were detected (Figure 5C, see legend for statistics). Although it is possible that this approach fragmented GFAP-positive branches, which would lead to an overestimation of the number of GFAP segments and an underestimation of their average length, this would apply to both experimental groups and would therefore not qualitatively affect the comparison between groups. We also further analysed the cross-section area of transduced EGFP-expressing astrocytes (data from Figure 4), which only revealed a tendency towards smaller territories expressing RhoA-WT (not illustrated, one-way ANOVA, *p* = 0.017; post hoc Tukey test, *p* = 0.091 for RhoA-WT and *p* > 0.35 for RhoA-CA and RhoA-DN). In summary, across experimental conditions in vitro and in vivo, astrocyte remodelling was most consistently induced by RhoA-CA expression. In vivo, RhoA-CA altered the morphology of fine but not large GFAP-positive astrocytic processes. 

## 4. Discussion

Our first goal was to develop a viral experimental approach to modify astrocytic RhoA activity in vitro and in vivo and to test its effect on astrocyte morphology. We chose three RhoA variants, overexpressed them virally in astrocytes in vitro and in vivo, and identified their effect on astrocyte morphology. We quantified changes of astrocyte morphology after 3 days in vitro and 2–4 weeks in vivo, which are typical experimental timelines to reach sufficient expression of the mutants and to test their effects. Only the expression of a constitutively active RhoA mutant (RhoA-CA) consistently changed the morphology of hippocampal astrocytes in vitro and in vivo, whereas overexpression of the wild-type RhoA (RhoA-WT) and particularly of a dominant-negative RhoA (RhoA-DN) had variable outcomes across preparations and morphological features. Regarding RhoA-DN, it should be noted that the T19N mutation significantly reduces the binding affinity of the RhoA activating guanine nucleotide exchange factors (GEFs) [30]. However, overexpression of RhoA-DN may at the same time lead to a higher basal RhoA activity compared to physiological RhoA levels, which could explain variable results between model systems. 

In contrast to acute manipulations of RhoA activity on a timescale of minutes, our findings are the result of prolonged manipulations of RhoA activity. Compensatory activation of other signalling pathways modulating RhoA and/or its downstream effectors (for instance ROCK, LIMK, and cofilin) [43,47] could counteract a prolonged change of RhoA activity. Indeed, signalling pathways via Rac1 and Cdc42 are known to interact with RhoA signalling and/or to control astrocyte morphology [23,25,47,48,49,50]. Similarly, signalling by second messengers such as cAMP and Ca^2+^ can control, and can be controlled by, astrocyte morphology [12,26,37,51]. In addition, we recently showed that astrocytic 5HT_4_Rs activate both Gα_S_ and Gα_13_, which on their own have opposite effects on the balance between F-actin and G-actin in cultured astrocytes and on their morphology [29]. Therefore, the observed effects of expression of RhoA variants are likely to reflect the perturbation of a complex signalling network controlling astrocyte morphology. 

### 4.1. Mechanisms Underlying RhoA-CA Effect on Astrocyte Morphology

Cultured astrocytes expressing RhoA-CA had a reduced area, a more roundish shape, and fewer and shorter processes (Figure 1 and Figure 2). In other words, they lost the so-called stellate morphology of the control cells. This is in line with previous studies showing that an increase of RhoA activity in astrocytes can prevent and reverse the process of stellation [24,25,26]. When astrocytes were transduced in vivo, the effect on their morphology was assessed in acute brain slices. We previously showed that morphological parameters of astrocytes like the volume fraction obtained from perfusion-fixed tissue and acute slices match when corrected for tissue shrinkage [42], which indicates that our measurements in acute slices are a valid readout of astrocyte morphology and its changes in vivo. We observed that astrocytes in vivo expressing RhoA-CA had lower overall volume fractions and, when divided into a grid of small local subregions, more subregions with a very low amount of local astrocytic volume (Figure 4). This is in line with a collapse or shrinkage of fine astrocytic processes [3,13]. In contrast, neither the cross-section area of astrocytes nor the structure of their larger GFAP-positive processes was affected, which indicates that RhoA-CA selectively altered the peripheral and small astrocyte processes in vivo.

These structural changes most likely involve the reorganisation of the actin cytoskeleton, which can be controlled by RhoA as reviewed previously [23,52] and as demonstrated in the current study in cultured astrocytes. For instance, acute inhibition of the Arp2/3 complex in culture and in acute brain slices led to morphologically less complex astrocytes, which had fewer small S100β-positive protrusions and additional bigger processes [27]. This is similar to the redistribution of astrocyte process volume that we observed after RhoA-CA expression (Figure 4C). In the same study [27], it was shown that Arp2/3 inhibition increased the amount of active GTP-bound RhoA and that the Arp2/3 effect depended on the Rho-dependent kinase (ROCK). Collectively, these and our results indicate that a reduced activity of Arp2/3 and increased activity of RhoA lead to morphologically less complex astrocytes at the level of their peripheral small processes. Consequently, astrocytes require a relatively low baseline RhoA activity to maintain their highly branched complex morphology.

RhoA-CA may also act primarily on small astrocyte processes because GTPases of the Rho family are important signalling molecules up and downstream of the actin binding protein ezrin, which links the cell membrane to the actin cytoskeleton [53,54]. In astrocytes, ezrin was primarily detected in the small peripheral astrocyte processes, where it is important for astrocyte process motility [55,56]. Therefore, RhoA-CA could modify the structure of small astrocyte processes through ezrin and how ezrin links the astrocyte membrane and the cytoskeleton.

In a previous study, we revealed that acute inhibition of cofilin phosphorylation induces an astrocyte process withdrawal on a time scale of minutes [13]. In our present experiments, RhoA-CA expression also led to a withdrawal of peripheral processes over days and weeks. However, according to the literature, an increase of RhoA activity should lead to increased cofilin phosphorylation [43,57]. A straightforward explanation is that RhoA has downstream effectors other than cofilin, such as mDia, the myosin II regulatory light chain phosphatase, and ezrin [54,57], and that cofilin is controlled by many pathways other than RhoA [47]. This indicates that a type of morphology change cannot be clearly attributed to the activity of one specific signalling pathway alone, that the duration of the experimental manipulations or of RhoA activity needs to be considered, and that further dissection of the molecular mechanisms that determine and maintain astrocyte morphology is needed.

### 4.2. Outlook

We have not fully resolved how RhoA-CA changes the exact spatial relationship between individual synapses and PAPs, which requires an in-depth analysis of single spines with suitable single-synapse functional readouts and super-resolution microscopy [21,58]. Nonetheless, astrocytic expression of RhoA-CA could be a useful tool for answering interesting open questions. For instance, astrocytic expression of RhoA-CA is likely to perturb the physiological relationship between excitatory synapse size and local glutamate uptake [21]. It is therefore an experimental tool for uncovering the functional relevance of this relationship (e.g., for excitatory synaptic transmission and its plasticity). Similarly, it will also alter the spatial relationship between GABAergic synapses and nearby astrocytic processes [58], which again is an opportunity for exploring the relationship between tripartite synapse structure and synapse function.

It has recently been shown that reducing the expression of the protein ezrin in astrocytes reduces their morphological complexity [59] and induces a withdrawal of PAPs from excitatory synapses [60], which is accompanied by an impaired performance in an object location memory test [59] and increased expression of contextual fear memory [60], respectively. It will be interesting to test if astrocytic expression of RhoA-CA leads to similar behavioural phenotypes. Finally, RhoA-CA could be a useful tool to further investigate the role of astrocytic RhoA in astrocyte reactivity in culture models [38] or after brain injury [28,61].

## Figures and Tables

**Figure 1 cells-12-00331-f001:**
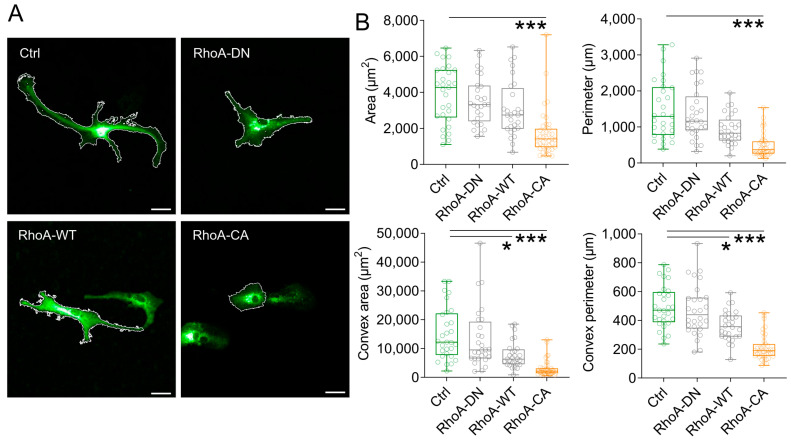
**Activation of the RhoA pathway changes astrocyte morphology in vitro.** (**A**) Increasing RhoA activity leads to retraction of astrocyte processes and a rounder cell shape. Representative images of astrocytes infected with rAAVs expressing variants of the small GTPase RhoA (dominant-negative, RhoA-DN; wildtype, RhoA-WT; constitutively active, RhoA-CA) tagged with mCherry for visualization or the far-red fluorescent protein TurboFP650 (Ctrl) under a mouse GFAP (mGFAP) promoter. White lines depict the region of interest (ROI) used for analysis (scale bars 20 µm). (**B**) Evaluation of astrocyte morphology shows a reduction of astrocyte area, perimeter, convex area, and the perimeter of the convex area with increasing RhoA activity (*p* < 0.001, Kruskal–Wallis test; Dunn’s multiple comparisons post hoc test, * *p* ≤ 0.05, *** *p* ≤ 0.001; cells from 4 individual cultures; n = 29, 28, 28 and 39 cells for Ctrl, RhoA-DN, RhoA-WT, and RhoA-CA, respectively). Box and Whisker plots: box indicates the 25th and 75th percentiles, the whiskers the minimum and maximum values, the horizontal line in the box the median, and hollow circles on top the individual data points.

**Figure 2 cells-12-00331-f002:**
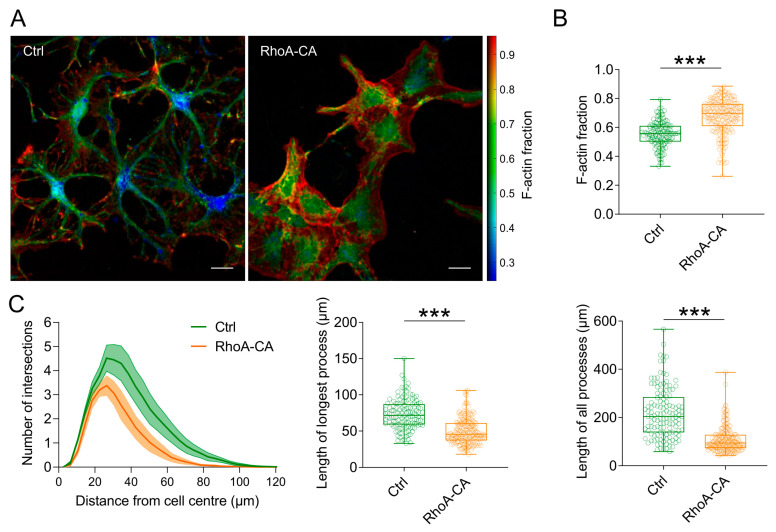
**Astrocytic expression of RhoA-CA increases the F-actin fraction in vitro.** (**A**) Representative images of cultured mouse hippocampal astrocytes expressing tdTomato (Ctrl, AAV-hGFAP-tdTomato) or RhoA-CA and tdTomato (AAV-hGFAP-tdTomato-RhoA-CA) stained for filamentous and globular actin structures (F-/G-actin). Displayed is the color-coded F-actin fraction (F-actin/(F-actin + G-actin)) (scale bars 20 µm). (**B**) Increased F-actin fraction in astrocytes expressing RhoA-CA (0.67 ± 0.12) compared to control cells (0.55 ± 0.09, Mann–Whitney U-test, *p* < 0.001). (**C**) Sholl analysis shows a reduced number of processes, defined by the number of intersections, in astrocytes expressing RhoA-CA. Area under the curve (AUC, left panel) was reduced in RhoA-CA (AUC 105 ± 8.4) compared to the control experiments (AUC 186 ± 12.1; unpaired Student’s *t*-test, *p* = 0.0015). Overexpression of RhoA-CA reduced the mean length of the longest processes (middle panel; Ctrl: 73.3 ± 20.6 µm; RhoA-CA: 49.6 ± 17.5 µm; Mann–Whitney U test, *p* < 0.001) and the total length of all processes (right panel; Ctrl: 219.0 ± 103.0 µm; RhoA-CA: 108.1 ± 52.4 µm; Mann–Whitney U test, *p* < 0.001). For (**B**,**C**), four independent cultures were analysed (Ctrl: n = 132 cells; RhoA-CA: n = 210 cells). Box and Whisker plots: box indicates the 25th and 75th percentiles, the whiskers the minimum and maximum values, the horizontal line in the box the median, and hollow circles on top the individual data points. *** indicates *p* ≤ 0.001.

**Figure 3 cells-12-00331-f003:**
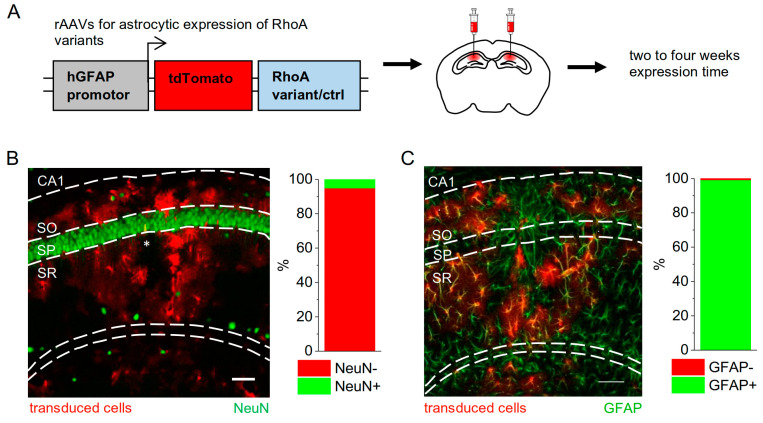
**Strategy and specificity for manipulating astrocytic RhoA activity in vivo.** (**A**) Schematic representation of the recombinant adeno-associated virus (rAAV) design. A human GFAP promotor was used to drive expression of the RhoA variants RhoA-WT/DN/CA fused with tdTomato or only of tdTomato (ctrl). rAAVs were injected into the dorsal hippocampus. Experiments were performed two to four weeks after virus injections. (**B**) Cellular specificity was tested by visualising transduced cells and neurons using immunohistochemistry. It was found that 94.9 ± 1.9% of the transduced cells were negative for the neuronal marker NeuN (NeuN+, n = 7 tissue slices from 3 animals, 977 cells in total). (**C**) The vast majority of tdTomato-expressing cells were positive for the astrocyte marker GFAP (99.2 ± 0.18%, n = 9 tissue slices from 3 animals, 2096 cells in total). Scale bars in (**B**,**C**) 100 µm. Note that data presented in (**B**,**C**) are from independent sets of immunohistochemistry experiments. * in (**B**) indicates the approximate virus injection site.

**Figure 4 cells-12-00331-f004:**
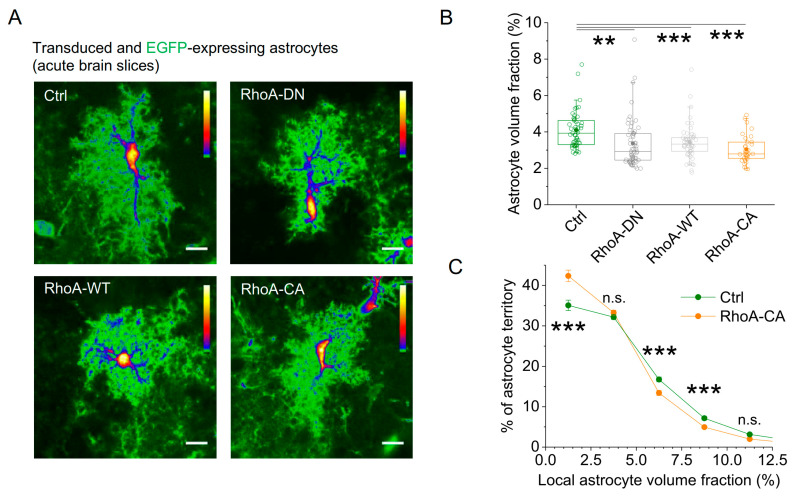
**Increasing astrocytic RhoA activity in vivo reduces the amount of fine astrocytic processes.** For testing the effect on astrocyte morphology, the four rAAVs were injected into dorsal hippocampi of mice in which astrocytes sparsely express EGFP. Astrocyte morphology was investigated in acute slices obtained from these mice by visualizing transduced EGFP-expressing astrocytes using 2PE fluorescence microscopy. (**A**) Sample images (tdTomato signal not illustrated, scale bar 10 µm). (**B**) The volume fraction (VF) of the astrocyte excluding the soma and major branches was used to quantify the effect of RhoA variants on astrocyte morphology. Expression of all RhoA variants significantly reduced the astrocyte VF (Kruskal–Wallis test, *p* < 0.001; Dunn post hoc tests, *p* < 0.01, <0.001 and <0.001 for RhoA-DN, RhoA-WT, and RhoA-CA, respectively, ** for *p* < 0.01 and *** for *p* < 0.001). (**C**) For analysing the subcellular effects of RhoA expression, individual astrocytes were divided into evenly spaced regions of interest (see Materials and Methods), and the distribution of subcellular volume fractions was determined for each cell. Average distributions are calculated and displayed for RhoA-CA and control rAAVs. They reveal a shift towards lower local volume fractions when astrocytes express RhoA-CA (two-way repeated measures ANOVA, *p* < 0.001 for Ctrl vs. RhoA-CA; post hoc Tukey test *** for *p* < 0.001 and n.s. for *p* > 0.90; *p* > 0.90 for local volume fractions > 12.5%). For (**B**,**C**), slices were analysed from at least 3 animals per condition (n = 51, 57, 51, and 33 cells for Ctrl, RhoA-DN, RhoA-WT, and RhoA-CA, respectively). SO, SP, SR represent *stratum oriens*, *stratum pyramidale*, *stratum radiatum*, respectively. Box and Whisker plots: box indicates the 25th and 75th percentiles, whiskers the 5th and 95th percentiles, the horizontal line in the box the median, the filled circle the mean, and hollow circles the individual data points.

**Figure 5 cells-12-00331-f005:**
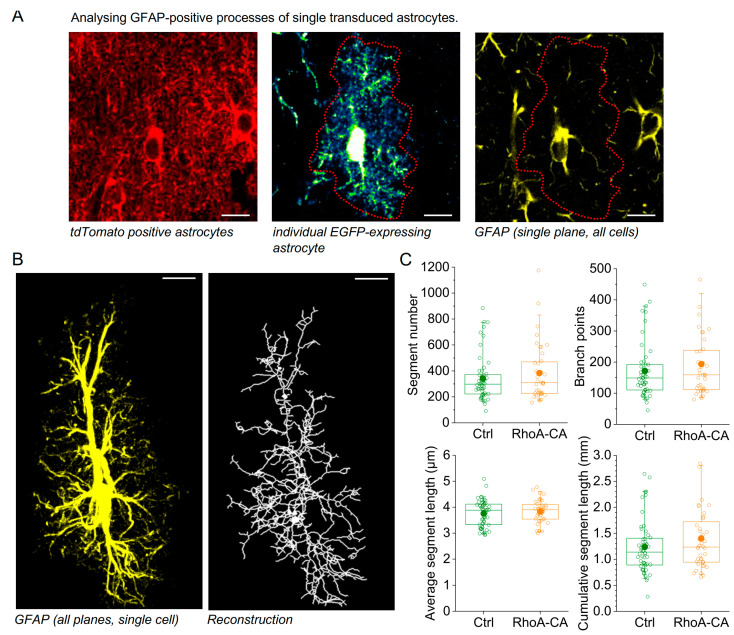
**Increasing astrocytic RhoA activity does not affect larger, GFAP-positive processes.** (**A**) Illustration of the experimental approach. Astrocytes in mice with sparse hippocampal EGFP expression were transduced using control and RhoA-CA–expressing rAAVs (hGFAP-tdTomato and hGFAP-tdTomato-RhoA-CA) (left panel). Isolated transduced and EGFP-expressing astrocytes were identified, and their boundaries were determined using their EGFP-fluorescence throughout an image stack (middle panel, single plane example, boundary indicated by dotted red line). Right panel: Example of GFAP immunohistochemistry of the section shown in the left and middle panel (single plane) illustrating GFAP staining of several astrocytes. The boundary of the single astrocyte selected for analysis is indicated by the dotted red line. (**B**) The GFAP immunofluorescence was masked outside of the single astrocyte under study (left panel, maximum intensity projection of the entire image stack). Right panel: reconstruction of GFAP-positive astrocyte processes obtained using MotiQ as described in [40]. (**C**) Numerical characterisation of GFAP reconstructions of 52 control and 40 RhoA-CA–expressing astrocytes from 4 and 3 virus-injected mice, respectively. Top left panel: number of GFAP segments (Mann–Whitney U-test, *p* = 0.46). Top right panel: number of branch points in the reconstruction (Mann–Whitney U-test, *p* = 0.47). Bottom left panel: average lengths of GFAP segments (Student’s *t*-test, *p* = 0.43). Bottom right panel: cumulative length of all GFAP segments per cell (Mann–Whitney U-test, *p* = 0.28). Box and Whisker plots: box indicates the 25th and 75th percentiles, whiskers the 5th and 95th percentiles, the horizontal line in the box the median, the filled circle the mean, and hollow circles the individual data points.

## Data Availability

The datasets supporting the current study have not been deposited in a public repository but are available upon request from the corresponding author.

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
