# Peer review of "Induced Remodelling of Astrocytes In Vitro and In Vivo by Manipulation of Astrocytic RhoA Activity"

_cells, 2023, doi:10.3390/cells12020331_

Round 1

Reviewer 1 Report

The manuscript Domingos et al. investigates how RhoA-related signalling controls astrocyte morphology.

Using a various transgenic animals and an AAV-based transfection strategy the authors investigated how overexpression of wild-type RhoA, of a constitutively active RhoA mutant (RhoA-CA), and a dominant-negative RhoA variant induced remodelling of cultured astrocytes. The authors report that increased Rho activity, particularly by the RhoA-CA variant, induces strong morphological changes in cultured astrocytes which are accompanied with robust alterations of the astrocytic actin assembly. Similarly, they find that increasing the astrocytic RhoA activity reduces the number of fine astrocytic processes in vivo. Consequently, the authors conclude, that astrocytes require low baseline RhoA activity in order to maintain their highly branched stellate morphology.

Overall, the study was performed at a technically sophisticated level and provides valuable quantitative data on astrocyte restructuring after manipulation of RhoA activity. However, despite the innovative nature of the author's experimental strategies, the work remains rather at the descriptive level and does not provide new insights into the actual mode of action of RhoA GTPases or the functional consequences of astrocytic restructuring. It is well known that Rho family of GTPases are key regulators of the actin cytoskeleton. Furthermore, downstream consequences of the RhoA pathway may include not only cytoskeletal reorganization, but also effects on gene expression and differentiation.

Given these limitations, I feel that the manuscript in its present form might be suitable for a more specialized journal.

Author Response

It is correct that our study is largely descriptive, which we, however, do not consider as a negative point. We would also like to point out that the predicted effect of RhoA-CA on the relative levels of filamentous and globular actin was confirmed in cultures (Fig. 2A-B) thus providing some mechanistic insights.

Maybe more important is the observation that expression of RhoA-DN and RhoA-WT did not have the expected effects on astrocyte morphology in both preparations. One expectation was that bi-directional morphology changes could be induced by RhoA-DN and RhoA-CA, which we now state more clearly, and which was, however, not observed. Our results also indicate that obtaining negative results using overexpression of wild-type GTPases of the Rho family or their manipulation (e.g. by pharmacology) can be misleading regarding the role of a specific GTPase. We extensively discussed possible reasons for that and consider this to be important beyond astrocytes.

It is also correct that the relationship between RhoA activity and astrocyte morphology has been studied before. However, most studies were performed exclusively on cultured astrocytes. We here demonstrate that in vivo RhoA-CA only affects the peripheral astrocyte processes and not the GFAP-positive larger processes. This is strikingly different compared to cultured cells, whose morphology is much more profoundly altered.

Reviewer 2 Report

The work submitted by Cátia Domingos and coauthors is devoted to such interesting and important topic as regulation of astrocyte morphology and its role in physiological processes. The work can be recommended for publication. There are only some points that should be discussed.  

 - In terms of the traditionally used protocols for preparation of primary astrocyte cultures, seeded cells should be grown in DMEM/FBS medium (10.1007/978-1-61779-452-0_4) to enhance glial cell proliferation. Neurobasal medium and especially B27 supplement were designed to promote neuronal survival and maturation (10.1002/jnr.490350513). Moreover, glial cell proliferation is attenuated in the case of Neurobasal/B27 combination. Why did the authors choose B27-supplemented medium for astrocyte culture preparation? Could DMEM/FBS medium influence the RhoA-mediated effects on astrocyte morphology in vitro? 

 - It is well known that different exposures results in astrocyte reactivation. Have the changes in RhoA activity impacted astrocyte phenotype in this case (reactive/non-reactive)?

Author Response

- In terms of the traditionally used protocols for preparation of primary astrocyte cultures, seeded cells should be grown in DMEM/FBS medium (10.1007/978-1-61779-452-0_4) to enhance glial cell proliferation. Neurobasal medium and especially B27 supplement were designed to promote neuronal survival and maturation (10.1002/jnr.490350513). Moreover, glial cell proliferation is attenuated in the case of Neurobasal/B27 combination. Why did the authors choose B27-supplemented medium for astrocyte culture preparation? Could DMEM/FBS medium influence the RhoA-mediated effects on astrocyte morphology in vitro? 

It is indeed well-established that the culturing protocol has profound effects on the morphology of astrocytes (see for instance Wolfes et al. 2016, Pirnat et al. 2021, Badia-Soteras et al. 2022 and references therein). Across the available literature, it seems that serum-containing media favour the development of polygonal morphologies. Instead, we aimed for a stellate morphology to be able to detect bi-directional structural changes, which we previously achieved using our culturing protocol (Müller et al. 2021). Attenuated glial cell proliferation was deemed beneficial because we prefer sparser cultures to simplify analysis. We have added additional information to the culturing section including the references mentioned above (page 4 of the revised manuscript) and refer to the role of culturing conditions for astrocyte morphology in the Results section (page 8 of the revised manuscript).

- It is well known that different exposures results in astrocyte reactivation. Have the changes in RhoA activity impacted astrocyte phenotype in this case (reactive/non-reactive)?

The question regarding RhoA activity and astrocyte reactivity is very interesting. In culture experiments, we have not systematically compared GFAP expression levels between the different experimental conditions. In vivo, we did explicitly analyze the morphology of GFAP-positive processes (Fig. 5). For RhoA-induced reactivity, we would expect an increase of segment numbers and probably also of segment length and branch points, because the increased GFAP expression improves detection of the processes. However, this was not observed, which indicated to us that RhoA-CA expression on its own does not induce astrocyte reactivity. Nonetheless, it may still modify astrocyte reactivity. We have expanded the very last statement of the Outlook section (page 14 of the revised manuscript).

Round 2

Reviewer 1 Report

The authors responded to my comments and agreed with most of the criticism.

Thus, the work remains more on the descriptive level. However, with further experiments, as suggested by me, the work could provide new insights into the actual mode of action of RhoA-GTPases or the functional consequences of astrocytic restructuring to increase the importance and visibility of the publication. In any case, I have no further objections to publication.

Reviewer 2 Report

All my comments have been addressed.